# Regulating synchronous oscillations of cerebellar granule cells by different types of inhibition

**Yuanhong Tang**[1], **Lingling An**[1,2]*, **Quan Wang**[1], **Jian K. Liu**[3,4]*

**1** School of Computer Science and Technology, Xidian University, Xi'an, China, **2** Guangzhou institute of technology, Xidian University, Guangzhou, China, **3** Centre for Systems Neuroscience, Department of Neuroscience, Psychology and Behaviour, University of Leicester, Leicester, United Kingdom, **4** School of Computing, University of Leeds, Leeds, United Kingdom

\* an.lingling@gmail.com (LA); jian.liu@leicester.ac.uk (JKL)

**Data Availability Statement:** The code used to generate the results in this paper is available on https://github.com/jiankliu/GC-CoC-Network.

**Funding:** This work was supported by National Natural Science Foundation of China Grant

## Abstract

Synchronous oscillations in neural populations are considered being controlled by inhibitory neurons. In the granular layer of the cerebellum, two major types of cells are excitatory granular cells (GCs) and inhibitory Golgi cells (GoCs). GC spatiotemporal dynamics, as the output of the granular layer, is highly regulated by GoCs. However, there are various types of inhibition implemented by GoCs. With inputs from mossy fibers, GCs and GoCs are reciprocally connected to exhibit different network motifs of synaptic connections. From the view of GCs, feedforward inhibition is expressed as the direct input from GoCs excited by mossy fibers, whereas feedback inhibition is from GoCs via GCs themselves. In addition, there are abundant gap junctions between GoCs showing another form of inhibition. It remains unclear how these diverse copies of inhibition regulate neural population oscillation changes. Leveraging a computational model of the granular layer network, we addressed this question to examine the emergence and modulation of network oscillation using different types of inhibition. We show that at the network level, feedback inhibition is crucial to generate neural oscillation. When short-term plasticity was equipped on GoC-GC synapses, oscillations were largely diminished. Robust oscillations can only appear with additional gap junctions. Moreover, there was a substantial level of cross-frequency coupling in oscillation dynamics. Such a coupling was adjusted and strengthened by GoCs through feedback inhibition. Taken together, our results suggest that the cooperation of distinct types of GoC inhibition plays an essential role in regulating synchronous oscillations of the GC population. With GCs as the sole output of the granular network, their oscillation dynamics could potentially enhance the computational capability of downstream neurons.

## Author summary

Synchronous oscillation dynamics is one of the ubiquitous features of neural population. It is thought to be coordinated by inhibitory neurons. However, there are various types of inhibition conveyed by different network pathways in neural circuits. Feedforward

62072355 (AL), Shanxi Key Research and Development Programs of China Grant 2019ZDLGY13-07 (QW), Zhejiang Lab of China Grant 2019KC0AB03 and 2019KC0AD02 (JL), and Royal Society Newton Advanced Fellowship of UK Grant NAF-R1-191082 (JL). The funders had no role in study design, data collection and analysis, decision to publish, or preparation of the manuscript.

**Competing interests:** The authors have declared that no competing interests exist.

inhibition is a pathway from external inputs to inhibitory neurons leading to excitatory neurons, whereas feedback inhibition is looped from excitatory neurons first and back to them via inhibition neurons. Moreover, gap junctions are frequently expressed between inhibitory neurons. It is far from clear that how neural synchronous oscillation dynamics is modulated by these different types of inhibition. Here we investigated this question using a computational model of the cerebellar granular layer network equipped with excitatory granular cells and inhibitory Golgi cells. We demonstrate that feedback inhibition is important to engender network oscillations, whereas gap junctions create robust oscillations. Further analysis of cross-frequency coupling between different frequency bands reveals that the coupling is regulated by Golgi cells through feedback inhibition. Together with current literature in research on the cerebellum, neural synchronous oscillation, and cross-frequency coupling, our results indicate that distinct types of inhibition could contribute to rich dynamics of the cerebellum and promote the cerebellum to play a functional character of cognition modulator in the communication with other brain areas.

## Introduction

The capacity of the brain to accommodate information precisely within a limited time window is important for reliable execution of complex sensory and motor behaviors [1, 2]. When the cerebellar cortex is injured, the movement loses precision and becomes uncoordinated and incorrectly timed, occurring in both humans [3–5] and animals [6, 7]. It indicates that the cerebellum has an extraordinary capability to accurately represent and process timing information during behaviors [8–10]. In the cerebellum, diverse types of signals of granular cells (GCs) through integrating information from mossy fibers (MFs) are critical for the function of the cerebellum [11], as them are converged to Purkinje cells (PCs) for fine-tuning of motor activities with tens of milliseconds [12].

One typical precise timing activity in neuron population dynamics is synchronous oscillation, in which neural populations fire spikes as a pacemaker in time. In contrast to oscillation at the level of single neurons, where intrinsic cellular properties, such as ion channels, play a dominant role [13, 14], synchronous oscillation at the network level involves synaptic interactions between neurons. Oscillatory network activity, as a ubiquitous feature of brain dynamics, are considered contributing to a variety of neural computations in sensory, motor, and cogitative behaviours [15–18]. Several mechanisms have been proposed for synchronous firing, including feedforward inhibition from inhibitory neurons to excitatory neurons [9, 19, 20], feedback inhibition where excitation dominates inhibition [21], opposite bursting patterns of excitatory and inhibitory neurons [22–25], and gap junctions between inhibitory neurons [26–30]. Furthermore, oscillations can also be modulated by the balance of excitation and inhibition in cells [31–34].

In the granular layer of the cerebellum receiving MF inputs, synchronous oscillations are exhibited in both GCs and Golgi cells (GoCs) [19, 35]. Previous studies have shown that feedforward inhibition (FFI) formed by GoCs to GCs provides a mechanism to produce information transmission precisely [9]. It also found that the emergence of oscillation is related to feedback inhibition (FBI) between GoCs and GCs [19, 36]. Moreover, gap junctions between GoCs can also generate oscillations in the GoCs network [37] and promote oscillations of GCs [35]. However, there is still a lack of understanding of how oscillation dynamics of GCs is regulated by a cohort of various types of inhibition together.

In this work, we addressed this question with a network model of GCs and GoCs equipped with different scenarios of inhibition, FBI and FFI between GCs and GoCs, and gap junctions between GoCs. Specifically, using MF-like Poisson spike inputs, we intend to explore how the synchronous oscillation of GCs is emerged and regulated through a cohort of inhibition, individually and altogether. We found FBI is crucial to the development of oscillations whereas regulated by FFI. However, the generation of GC oscillations requires an appropriate level of excitation and inhibition combination in synapses. Particularly, excitation and inhibition have contrary effects on the oscillation frequency to make it rise with increasing excitability, and decreases as increasing inhibition. Moreover, oscillations can be changed oppositely by two mechanisms: short-term plasticity of GoC-GC synapses suppressing oscillation, whereas gap junctions between GoCs promoting oscillation. Finally, we note that there is a significant level of cross-frequency coupling between slow and fast oscillations in GCs. Such a tight coupling was modulated by GoCs through feedback inhibition. Altogether, our results suggest that the interaction of various types of inhibition in neural networks contributes to regulating synchronous network oscillations. In the granular network of cerebellum, GCs are output neurons sending their dynamics to downstream neurons. Tight and rich oscillatory activity of GCs could potentially perform a functional role in developing, maintaining, and propagating information to downstream neurons of the cerebellum, as well as regulating the communication with other brain areas.

## Methods

### Single cell models

Granule cells (GCs) were modeled as integration-and-fire neurons as previously [38], whose membrane potentials $V$ obey the equation:

$$C_m \frac{dV}{dt} = -g_l(V - E_L) \exp\left(-(V - E_L)/5\right) - I_{noise} - g_{AHP}z_{AHP}(t)(V - E_K) - I_{syn}(t) \quad (1)$$

where $C_m$ is the membrane capacitance, $g_L$ is the leak conductance, and $E_L$ is the leak resting potential. When the membrane potential reaches the threshold $V_T$ at the spike time $t_{spk}$, $V$ was set to 40 mV for a duration of spike as $\tau_{dur} = 0.6$ ms. After spike, at $t = t_{spk} + \tau_{dur}$, the repolarizing potential was set to $V_{rest}$, and an afterhyperpolarization (AHP) conductance was activated. The gating variable $z_{AHP}$ followed the dynamics $dz_{AHP}/dt = (1 - z_{AHP})/x_{AHP} - z_{AHP}/\tau_{AHP}$. The resource variable $x_{AHP}$ obeyed the dynamics $dx_{AHP}/dt = -x_{AHP}/\tau_{AHP_x} + \delta(t - t_{spike} - \tau_{dur})$, where $\tau_{AHP_x} = 1$ ms. The refractory period was set as $t_{ref} = 2$ ms. To mimic the ongoing activity in the simple point neuron model, we injected a noisy excitatory current $I_{noise} = (V - V_E)g_N$ with a slowly fluctuating conductance $g_N$ described by an Ornstein-Uhlenbeck process, $\tau_N dg_N/dt = -g_N + \sigma_N\sqrt{\tau_N}b(t)$, where $\sigma_N = 0.12$ nS, $\tau_N = 1000$ ms, and $b(t)$ as white noise with unit variance density.

Golgi cells (GoCs) were modeled similarly as before [37], in which the membrane potential $V$ followed the equation:

$$C_m \frac{dV}{dt} = -g_l(V - E_L) - I_{Na} - I_{gap} - I_{noise} - g_{AHP}z_{AHP}(t)(V - E_K) - I_{syn}(t) \quad (2)$$

where the sodium current was given by $I_{Na} = g_L\Delta T \exp((V - V_T)/\Delta T)$ with the firing threshold $V_T$ and $\Delta T = 3$ mV. When the membrane potential reached $V_T$ at the spike time $t_{spk}$, $V$ was set to 40 mV for a duration of spike as $\tau_{dur} = 1$ ms. AHP current and noise current used the same form as in the GC model. The refractory period was set as $\tau_{ref} = 2$ ms. The current induced by gap junction in the $i^{th}$ cell was $I_{gap,i} = \sum_j g_{ij}(V_i - V_j)$, where $g_{ij}$ was the gap conductance between

**Table 1. The parameters of single cell models.**

| Neuron | C(pF) | $g_L$(nS) | $E_L$(mV) | $V_T$(mV) | $V_{rest}$(mV) | $g_{AHP}$(nS) | $E_K$(mV) | $\tau_{AHP}$(ms) |
|--------|-------|-----------|-----------|-----------|----------------|---------------|-----------|------------------|
| GoC | 20 | 1 | -50 | $-45 \pm 2.25$ | -50 | 4 | -100 | 20 |
| GC | 4.9 | 1.5 | -90 | $-49 \pm 2.45$ | -65 | 1 | -90 | 3 |

the $i^{th}$ and $j^{th}$ cells, $V_i$ and $V_j$ were membrane potentials of the $i^{th}$ and $j^{th}$ cells. The gap junction conductance was distributed based on the distance between GoCs and sampled according to experimental data as described previously [37, 39]. All the parameters of GCs and GoCs have the same values, except those listed in Table 1.

## Synapse models

For synaptic currents, the $I_{syn}$ of GCs represents the excitatory input from MFs and inhibition input from GoCs. The $I_{syn}$ of GoCs receives excitatory input from GCs and MFs. All the synaptic currents were modeled with a similar form as:

$$I_{syn} = g_{max}r(t)Y(V - E_{syn}) \tag{3}$$

where the scaling factor $Y$ was a voltage-dependent function for NMDA: $Y = 1/(1 + exp(-(V - 84)/38))$ and $Y = 1$ for other receptors. The gating variable $r$ was described by

$$r' = -r/\tau_{decay} + \alpha.s(1 - r)$$

$$s' = -s/\tau_{rise} + Ru\sum_k \delta(t - t_{spk}) \tag{4}$$

Short-term synaptic plasticity (STP) was modeled with a simple phenomenological model that describes the kinetics of such plasticity. It treats short-term depression and facilitation as two independent variables, $R$ and $u$, respectively [40, 41] as

$$R' = (1 - R)/\tau_{rec} - RU\delta(t - t_n)$$

$$u' = (U - u)/\tau_{fac} + U(1 - u)\delta(t - t_n) \tag{5}$$

We used four types of synaptic connections between neurons: excitatory MF-GC, MF-GoC, and GC-GoC synapses, and inhibitory GoC-GC synapses. Feedforward pathway MF-GoC-GC consists of MF-GoC synapses with slow and fast AMPA ($\alpha$-amino-3-hydroxy-5-methyl-D-aspartate) receptors, and GoC-GC synapses with slow and fast GABA (gamma-aminobutyric acid) receptors. Feedback pathway MF-GC-GoC-GC includes MF-GC synapses with slow, fast AMPA and NMDA (N-methyl-D-aspartate) receptors, GC-GoC synapses with fast AMPA receptor, and the same type of GoC-GC synapses. Synaptic parameters for each type of synapse were constrained by experimental data [9, 42, 43]. In those simulations with no STP included for GoC-GC synapses, variables R = 1 and u = U were held fixed without dynamic updates.

Synaptic delays in all synapses were included except GoC-GC synapses. The values of the delay were sampled from a Gaussian distribution (mean as 1 ms and standard deviation as 0.2 ms). The values of these various parameters are provided in Table 2. In this work, the strengths of excitation and inhibition are the main factor in the generation of oscillation. By default, we used $W_{MF-GC}$ = 3 nS, $W_{MF-GoC}$ = 3 nS, $W_{GC-GoC}$ = 3 nS and $W_{GoC-GC}$ = 4 nS for simulations without gap junction. When gap junctions were included in GoCs introducing additional inhibition, we used $W_{GoC-GC}$ = 2.5 nS. These values in the model are the means of synaptic weights, and sampled from a Gaussian distribution with a variance proportional to the mean.

**Table 2. The parameters of synapses.**

| Synapse | | Strength | Synaptic dynamics | | | Short-term plasticity | | |
|---|---|---|---|---|---|---|---|---|
| Pre-Post | type | $g_{peak}$(ns) | $\alpha$(1/ms) | $\tau_{rise}$(ms) | $\tau_{decay}$(ms) | U | $\tau_{rec}$(ms) | $\tau_{fac}$(ms) |
| MF-GC | AMPA$_{fast}$ | $W_{MF-GC}$ | 3 | 0.3 | 0.8 | 0.5 | 12 | 12 |
| | AMPA$_{slow}$ | $2 * W_{MF-GC}$ | 0.3 | 0.5 | 5 | 0.5 | 12 | 12 |
| | NMDA | $2.4 * W_{MF-GC}$ | 0.35 | 8 | 30 | 0.05 | - | - |
| MF-GoC | AMPA$_{fast}$ | $W_{MF-GoC}$ | 3 | 0.3 | 0.8 | 0.5 | - | - |
| | AMPA$_{slow}$ | $2 * W_{MF-GoC}$ | 0.3 | 0.5 | 5 | 0.5 | - | - |
| GC-GoC | AMPA$_{fast}$ | $W_{GC-GoC}$ | 3 | 0.3 | 0.8 | 0.5 | - | - |
| GoC-GC | GABA$_{fast}$ | $W_{GoC-GC}$ | 3 | 1 | 5 | 0.5 | 400 | 20 |
| | GABA$_{slow}$ | $0.15 * W_{GoC-GC}$ | 0.35 | 5 | 100 | 0.05 | 20 | 400 |

The values of synaptic weights listed here were used as the default mean values in our simulations, unless those were mentioned differently in the results below.

## Network model

The network model was set up with 500 MFs, 2000 GCs and 144 GoCs, where each GC received synaptic inputs from 4 MFs and 10 GoCs randomly, each GoC received inputs from 10 MFs and 50 GCs randomly. When there was no gap junction between GoCs, GCs and GoCs were arranged in a random network without specific network topology, and the synapses between cells were randomly selected for connections between GCs and GoCs in three types of network structures: feedforward inhibition, feedback inhibition, and both included.

When gap junctions were included and since the strength of gap junctions depends on the distance between GoCs, GoCs were arranged in a 2D grid space of 400 by 400 $\mu m$ to compute the distance between cells shown in S7 Fig. We used the same network topology and parameters for GoCs as in the previous experimental study [37]. Briefly, the locations of GoCs were arranged in a grid space where the position of each cell was randomly shifted with a small amount around the center. GoCs were arranged on a 12x12 grid with 33 $\mu m$ spacing. The radius of each cell was drawn randomly from 0.7 to 1.3 times of the average radius (70 $\mu m$). The conductance value of gap junction between two GoCs was taken proportional to the area of overlap between the two disks of both cells. We also distributed GCs randomly in this network of GoCs, whereas the synaptic connections between GCs and GoCs were still randomly selected from cells without the effect of distance between cells.

## Data analysis

The network was stimulated by a sequence of Poisson spikes representing the MF input to trigger GCs and GoCs. Simulations were run in C++ with a time step of 0.1 ms. Data collected after simulation was saved for further analysis. In order to study the impact of three types of network structures on GC firing activities, we simulated the network with 50 trials of 10 second each. The population of GC spikes averaged over all cells in all 50 trials was used to perform principal component analysis to obtain three clusters. The k-means clustering method was used to identify clustered groups in principal components. We computed the interval-inter spike (ISI), coefficient of variation (CV), standard deviation (SD) of spike trains of each cell, then averaged over the population for each trail. The number of spikes was counted during 1 second and averaged over all cells for each trial. These four measures were used as metrics of neural firing [44] for each type of GoC inhibition in the network. The same analysis was

performed in S7 Fig in order to determine the effect of gap junctions on top of different inhibition of network structures.

The frequency spectrum of firing rate over 10 seconds was obtained by Fast Fourier Transform using fft function in MATLAB. The network oscillation frequency (OF) was considered as existing if the power spectrum has a peak amplitude between 5 Hz and 200 Hz, and was defined as the frequency corresponding to the peak power in the power spectrum. To further analyze oscillation, we recorded the amplitude as the peak of firing rate, inter-event interval (IEI) as the distance between two peaks, and width as the distance between the points where the half amplitude reaches before and after the peak.

To characterize the population activity of GCs, we used filters with different cutoff frequencies implemented using fifth-order Butterworth filters in MATLAB. For cross-frequency coupling, we quantified the strength of phase-amplitude coupling using the Envelope-to-Signal Correlation (ESC) measure [45]. The ESC quantifies the strength by calculating the correlation between the amplitude envelope of the band-pass filtered high frequency signal and the band-pass filtered low frequency signal. Morlet wavelet filter width was set as 7, FFT-size as 200, shuffling windows as 200, and the sampling frequency as 500 Hz. We used a wide range of frequencies for low-pass (0–30 Hz) and high-pass filters (0–100 Hz) to compute the ESC for each pair of low and high frequency, which resulted in the plots in a 2D space [46].

## Results

We employed a neural network with 500 MFs, 2000 GCs and 144 GoCs, where MFs, GCs, and GoCs were modeled as modified integration-and-fire neurons yet constrained by experimental data as previously [37, 38]. Each GC received synaptic inputs from 4 MFs and 10 GoCs randomly, and GoC received inputs from 10 MFs and 50 GCs randomly. Synapses from MF to GC and GoC and from GC to GoC are modeled as excitatory APMA and NMDA dynamics while those from GoC to GC are inhibitory GABA dynamics [9, 38, 42, 43]. To account for the random effect of neural dynamics, a noisy current was injected into each neuron. To further incorporate heterogeneity in the network, we included a large amount of variation in neural and synaptic parameters by randomly sampling the values around the means. We modeled network input as MF spiking activity represented as Poisson spikes of 25 Hz, unless different frequencies were used in the cases mentioned. The code of our model is publicly deposited and the detail of the model can be found in Methods.

### Network synchronous oscillation induced by feedback inhibition

To investigate the impact of different scenarios on the activity of GCs, we considered a network model with three types of network connections illustrated in Fig 1A: FFI—the feedforward inhibition MF-GoC-GC pathway, where MFs excite GoCs that then inhibit GCs; FBI—the feedback inhibition pathway through MF-GC-GoC-GC synapses, where MFs trigger GCs that excite GoCs, then in turn, GoCs send inhibition back to GCs; FFI+FBI—a more general case where both FFI and FBI are undertaken. We then employed Poisson MF-like spike inputs to examine response patterns of GCs and GoCs under different scenarios. With the same stimulus, we found that GCs show a variety of responses caused by different formats of GoC inhibition in the network. Fig 1B shows example membrane potential traces of a single GC and GoC under three schemes. Not surprisingly, FBI decreases GC activity. If FBI paired with FFI, additional inhibition reduces GC spikes further. Detailed characteristics of spike responses show GC and GoCs have significantly varying firing activity due to different inhibition (S1 Fig).

One unique feature of network activity at the population level is the synchronous oscillation dynamics, where precise synaptic integration allows cells to fire within a restricted time

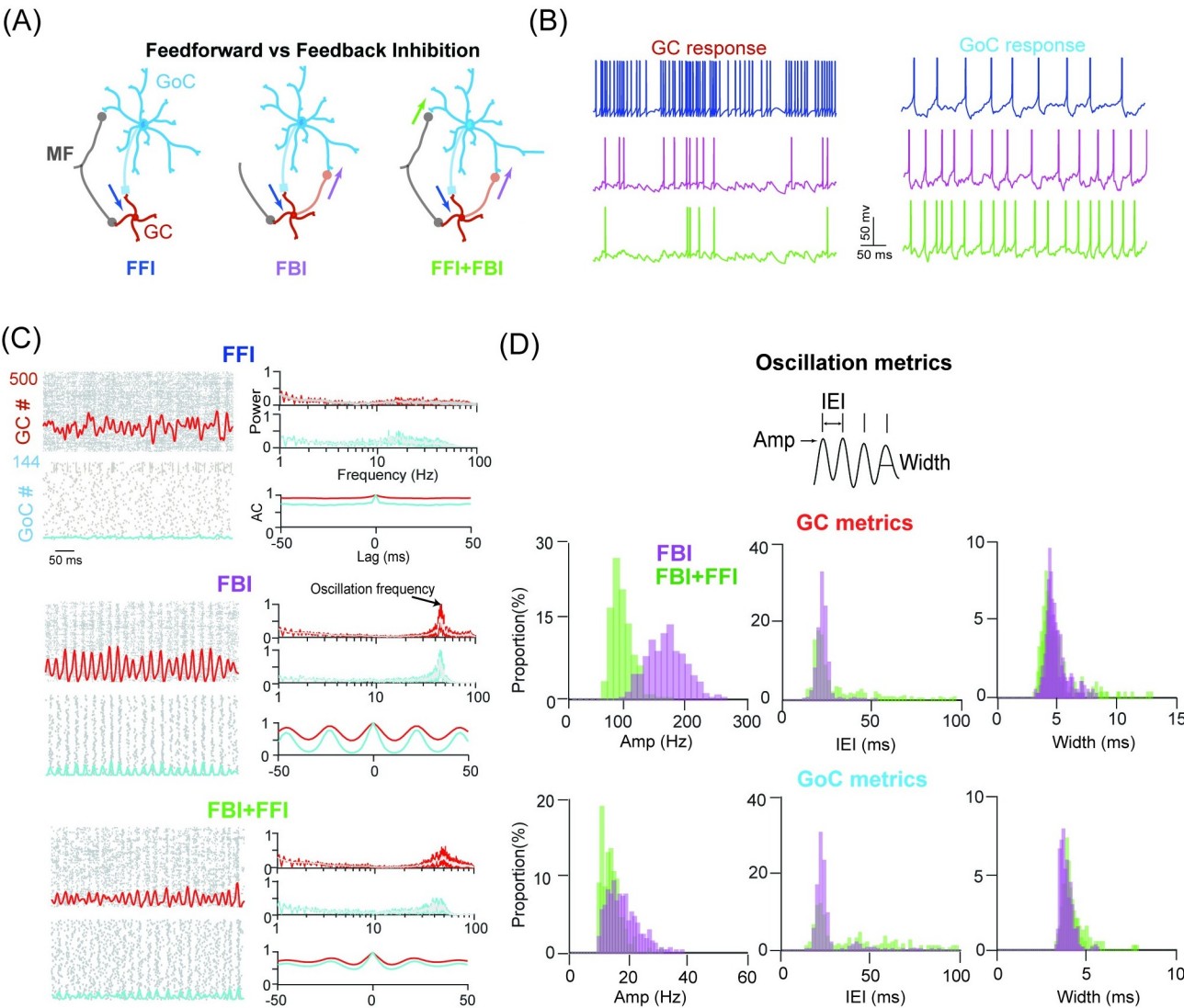

**Fig 1. Network synchronous oscillation induced by feedback inhibition.** (A) Schematic illustration of three scenarios of network connections between GCs and GoCs with feedforward inhibition (FFI, left), feedback inhibition (FBI, middle) and both inhibition loops (FFI+FBI, right). Mossy fiber (MF). Granule cell (GC). Golgi cell (GoC). (B) Example membrane potential traces of a single GC in three scenarios, evoked by 50 Hz Poisson MF inputs. (C) Spike raster and population firing rate of 500 GCs and 144 GoCs (left), and the corresponding power spectrum and autocorrelation (AC) of population firing rate (right) in three network scenarios. Note the network oscillation is present only with the FBI included. The peak of the power spectrum indicates the oscillation frequency of the network. (D) Network oscillation metrics characterized by the amplitude (Amp), inter-event interval (IEI) and width. Histograms of three metrics for GC (top) and GoC (bottom).

window. It has been shown that the synchronous oscillation in GCs can be induced by GoC inhibition [19]. Here we examined how GoC inhibition produces the GC oscillation with different network connections. We fixed the strengths of excitation and inhibition to GCs by setting synaptic parameters as $W_{MF-GC}$ = 3 nS and $W_{GoC-GC}$ = 4 nS, then studied the effect of different formats of GoC inhibition on oscillation dynamics. We found that network activity largely depends on the details of GoC inhibition (Fig 1C). Only when the FBI was included, there is a considerable level of oscillation seen by peaks in the power spectrum of the population firing activity of GCs and GoCs. FFI alone is not sufficient to induce oscillation. The indication of oscillation can also be observed using the autocorrelation of firing activity that

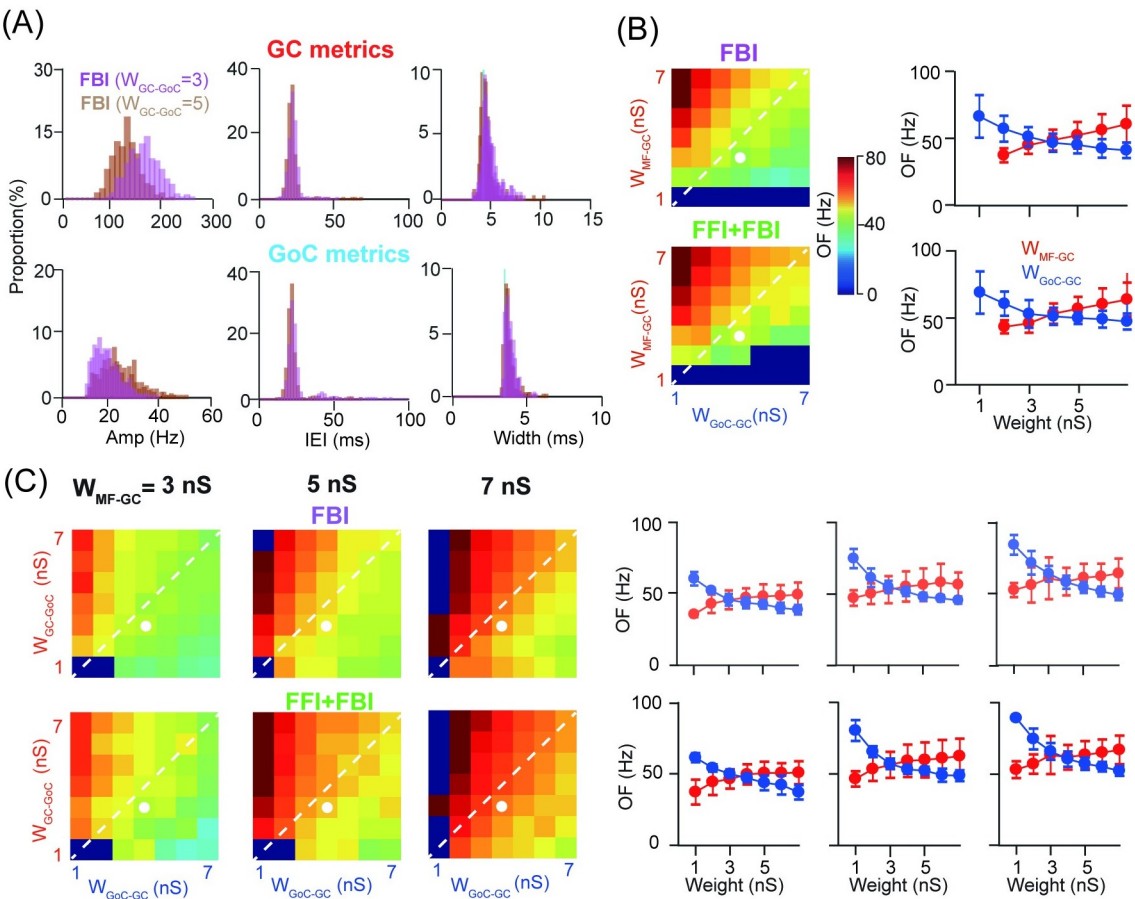

**Fig 2. Oscillation frequency controlled by excitation and inhibition strength.** (A) Example oscillation metrics changed by the excitation strength of GC-GoC synapses. (B) Modulation of oscillation frequency (OF) over increasing $W_{MF–GC}$ and $W_{GoC–GC}$ strengths under two network scenarios (left). The white dash line indicates the ratio of excitation and inhibition as 1. The point marked as white is the default values $W_{MF–GC}$ = 3 nS and $W_{GoC–GC}$ = 4 nS fixed for showing the monotonic change on the right plot. (Right)Example traces showing monotonic changes of the OF, with increasing excitation (blue) or inhibition (red) while fixing the other one along the default values. (C) Similar to (B), but in the parameter space of increasing $W_{GC–GoC}$ and $W_{GoC–GC}$ strengths, with three cases of $W_{MF–GC}$. The point marked as white is the default values $W_{GC–GoC}$ = 3 nS and $W_{GoC–GC}$ = 4 nS fixed for showing the monotonic change on the right plot.

displays rhythms with FBI, but not with FFI. Compared to the FBI, including FFI disturbs and weakens network oscillations. To characterize oscillation in detail, we introduced three measures, the amplitude, inter-event interval (IEI) and width of oscillation, as shown in Fig 1D. Adding FFI on top of FBI decreases the amplitude and narrows the distribution for GCs, as a result of the weak oscillation. In addition, both the IEI and width were increased slightly. These observations are also visible in the GoC population, except that the amplitude was mildly decreased, compared to GCs. These results show that the feedback inhibition of GoCs has a great influence on the GC oscillation.

To investigate whether oscillations could be impacted by different levels of inhibition, we changed the strength of GC-GoC synapses as part of FBI to increase the excitability of GoCs and make them easy to fire more spikes. However, there is no significant change in network oscillations (Fig 2A). More excitation from GC to GoCs can cause the amplitude of GC oscillation smaller and that of GoC larger, but there is little change on the IEI and width of oscillation that are more relevant to the frequency of the oscillation. To systematically examine the effect

of excitation and inhibition on the GC oscillation, we used a wide range of the parameters of synaptic weights and explored how the network oscillation frequency (OF), e.g., the frequency at which the power spectrum is peaked (indicated by the arrow in Fig 1A, middle), is changed by these parameters. Compared to FBI alone, adding FFI has no significant effect on the OF within a large range of excitation $W_{MF-GC}$ and inhibition $W_{GoC-GC}$ (Fig 2B). Interestingly, the OF is still changed as a balance of excitation and inhibition on GCs (indicated by the white line in Fig 2B). At fixed weights of excitation and inhibition, the change of OF is monotonic along each dimension of excitation or inhibition. Moreover, excitation and inhibition affect the OF oppositely: excitation promotes OF but inhibition suppresses it.

As noted above, changes in the GoC excitability can modulate network oscillation. However, these changes lead to a limited change of the OF (different levels of $W_{GC-GoC}$ in Fig 2C), compared to significant changes of the OF due to excitation of GCs at MF-GC synapses (different levels of $W_{MF-GC}$ in Fig 2C). Nevertheless, oscillation can only emerge in the network with enough excitation on GCs, whereas the balance of excitation and inhibition on GCs are required to maintain network oscillations. Further detailed analysis reveals that the oscillation of GCs is closely related to the balance of excitation and inhibition on GCs (S2 Fig). Because diverse responses can arise from combining different synaptic strengths, more systematic analysis of parameters in all types of synapses confirms the GC oscillation was mainly regulated by the balance of excitation and inhibition on GCs (S3 Fig). Overall, the modulation of MF-GC strength changes the GC excitability, and the modulation of MF-GoC and GC-GoC synaptic weights changes the GoC excitability. As a result, sufficient MF-GC excitatory inputs are a prerequisite to ensure the generation of oscillation. These results indicate the network oscillation can be induced by the GoC feedback inhibition, which is the key connection structure for significant oscillations of GCs.

## Network oscillation depressed by short-term plasticity of GoC-GC synapses

Oscillations require precise synaptic integration to allow neurons to fire within a confined time window. However, the nonlinear characteristic of short-term plasticity (STP) disrupts precise synaptic integration, produces jitters to spike time, and depresses the synchrony of the network activity [47] (see S4 Fig). Here we modeled the STP using a classic phenomenological model [40, 41] (see Methods) and installed it on GoC-GC synapses showing both facilitation and depression with repeated incoming spikes (S5 Fig). Indeed, we found that network oscillations of GC responses were distorted by the STP of GoC-GC synapses in feedback inhibition networks (Fig 3).

There are still significant peaks around 40 Hz in the power spectrum of GC population activities when the FBI was included (Fig 3A). Consistent with the results without STP, oscillations of GCs and GoCs are more notable in networks with FBI only, and are largely suppressed after pairing FFI with FBI. Moreover, the strength of oscillation is weaker than that without STP. The characteristics, amplitude, IEI and width of oscillation, show that oscillations were suppressed due to jitters of spike timing induced by STP, such that the amplitude decreases and both the IEI and the width becomes larger (Fig 3B). With a wide range of excitation and inhibition, the oscillation frequency was significantly reduced and restricted to a narrow range of synaptic parameters (Fig 3C). The depression due to the STP of GoC-GC synapses is robustly observed under a wide range of the STP parameters, independent of detailed profiles of facilitation or depression (S5 Fig). However, these changes in oscillations were little impaired by the STP of excitatory MF-GC synapses (S6 Fig). These results suggest that the STP of feedback GoC-GC synapses can greatly deteriorate network oscillations.

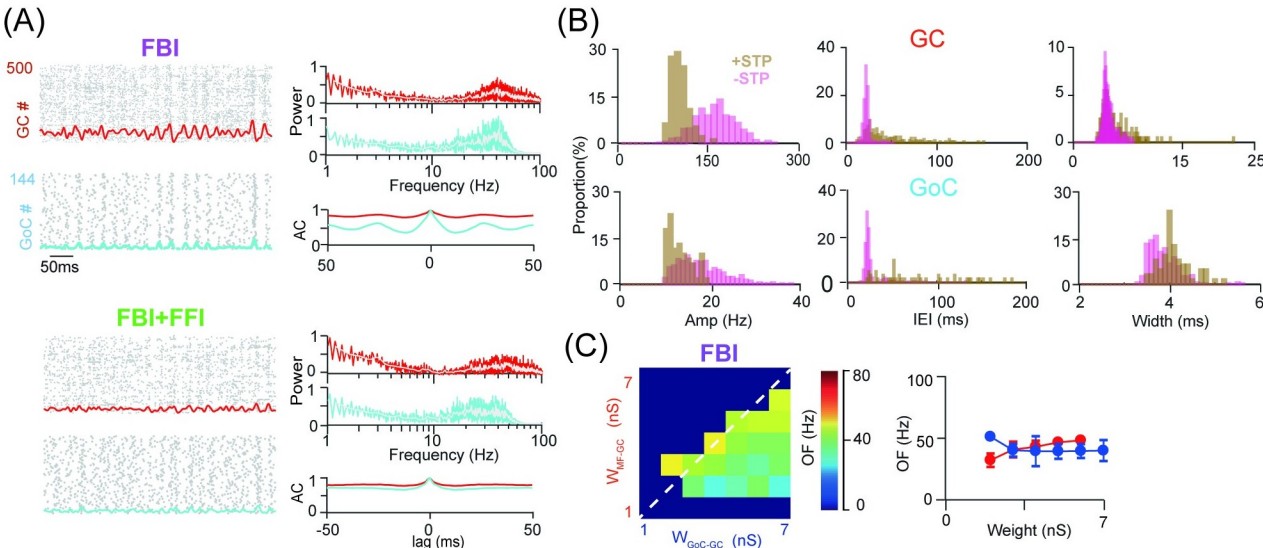

**Fig 3. Oscillation suppressed by short-term plasticity of GoC-GC synapses.** (A) Spike raster and population firing rate of all GCs and GoCs (left), and the corresponding power spectrum and auto-correlation (AC) of population firing rate (right) in two network scenarios of FBI (top) and FBI+FFI (bottom). Here short-term plasticity (STP) of GoC-GC synapses was included. (B) Oscillation metrics characterized by the amplitude (Amp), inter-event interval (IEI) and width. Histograms of three metrics for GCs (top) and GoCs (bottom) with STP (+STP) and without STP (-STP). (C) Modulation of the oscillation frequency (OF) over increasing $W_{MF-GC}$ and $W_{GoC-GC}$ strengths in networks with FBI. The point marked as white the left plot is the default values $W_{MF-GC}$ = 3 nS and $W_{GoC-GC}$ = 4 nS fixed for showing the monotonic change on the right plot.

## Network oscillation reinforced by gap junctions between GoCs

Previous findings suggest that GoCs play a vital role in modulating the GC activity [48], especially gap junctions between GoCs can generate and promote the GC synchronization [37]. Here we evaluated whether the GC activity still depends on different network scenarios of inhibition when gap junctions are included in GoCs. Since gap junctions are associated to the distance between cells, our network model has a layout of 2D grid space similar to the previous study [37] constrained by experimental data, where both GCs and GoCs were placed on grids but with random shifts, and the weights of gap junctions were sampled according to the distance between GoCs (S7 Fig, see Methods).

With gap junctions installed in GoCs, GC responses have different firing patterns under three scenarios of inhibition. More importantly, as the STP of GoC-GC synapses can suppress oscillation, hereby, we considered how gap junction, on top of the effect of STP, can affect oscillation. Consistent with the previous modelling study and experimental data that gap junction can generate the GoC synchronization and oscillation [37], we found that, when there was no feedback from GoCs, GCs exhibited changing patterns of pulsating rhythms that were induced by low-frequency GoC oscillations (Fig 4A, FFI). When GoC feedback inhibition was included in GCs, one can observe typical oscillatory activities with higher frequencies (Fig 4A, FBI and FBI+FFI). Notably, the dynamics of the GC population was mainly regulated by GoCs, particularly, by the synchronization of GoCs from gap junctions.

Compared to the network without gap junction, network oscillations are more prominent, as seen from the power spectrum and auto-correlation of population activity (Fig 4A, right), and three characteristics of oscillation (Fig 4B) in both GCs and GoCs. Thus, gap junction can overcome the distortion of GoC-GC synaptic STP to make oscillation more noticeable. Network oscillations induced by gap junction were robust to the change of the strength of gap junction (S8 Fig). When GoC-GC STP was switched off and only gap junctions were used, the

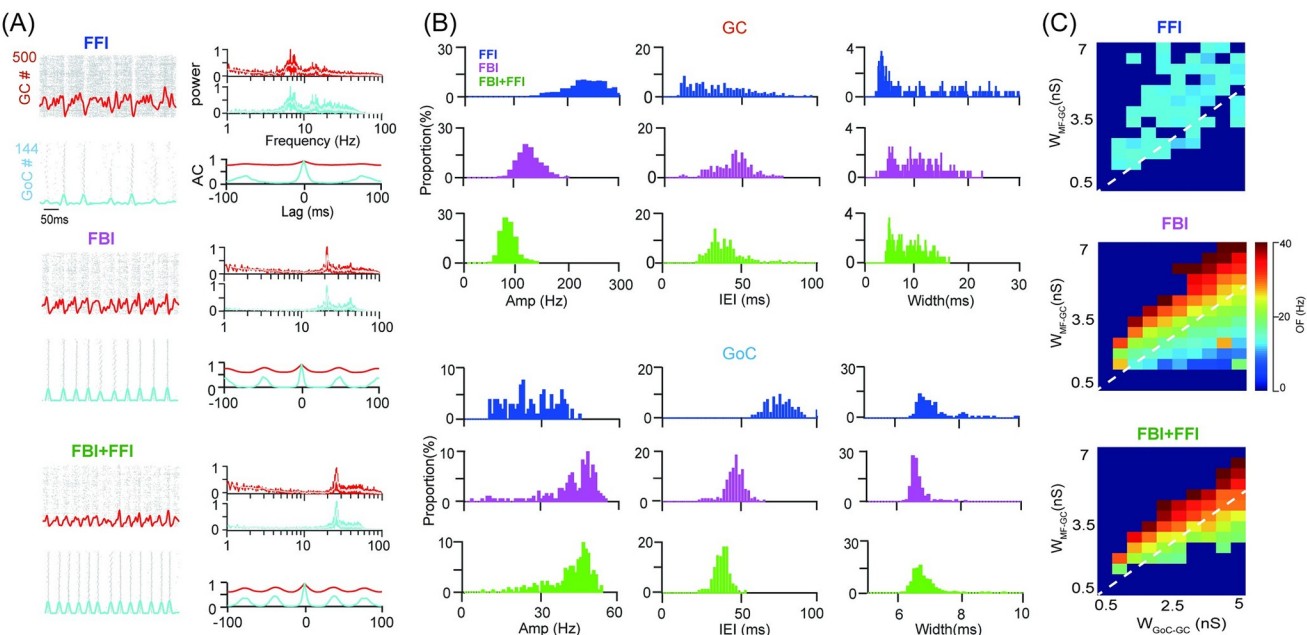

**Fig 4. Oscillation reinforced by gap junction.** (A) Spike raster and population firing rate of GCs and GoCs (left), and the corresponding power spectrum and auto-correlation (AC) of population firing rate (right) in three scenarios with gap junction and GoC-GC STP included. (B) Oscillation metrics characterized by the amplitude (Amp), inter-event interval (IEI) and width. Histograms of three metrics for GC (top) and GoC (bottom). (C) Modulation of the oscillation frequency over increasing excitatory (E) $W_{MF–GC}$ and inhibitory (I) $W_{GoC–GC}$ strengths. The white dash line indicates the ratio of excitation and inhibition as 1.

network shows more notable oscillations (S9 Fig). The oscillation frequency over a range of excitation and inhibition onto GCs shows that the FFI generates weak oscillations, compared to the FBI. Interestingly, oscillations were more tightly balanced by excitation and inhibition inputs on GCs via gap junctions of GoCs (the OF aligns with diagonal lines in Fig 4C), compared to the networks without gap junction (Fig 2).

More systematic analysis of the variety of strengths of all types of synapses confirms that the oscillation of the GC population via gap junction was in line with the balance of direct excitation from MFs and inhibition from GoCs onto GCs (S10 Fig). These results suggest that, although different network connections between GoCs and GCs, in particular, feedback inhibition, can generate network oscillation in GCs, gap junction between GoCs plays a functional role in robust oscillations. Moreover, gap junctions are able to tune excitation and inhibition converged to GCs and make them more balanced during oscillations in the network. Therefore, different types of inhibition, e.g. gap junctions together with feedback inhibition, take action in a cohort to regulate network oscillation more efficiently and effectively.

## Cross-frequency coupling of oscillations

The oscillatory neural activity, as a common feature of brain dynamics, has been observed in different frequency bands, from slow $\theta$ to fast $\gamma$ waves. The above results demonstrate there is a wide range of oscillation frequencies in the population activity of GCs. One of the intrigue features of rhythms, particularly in the brain dynamics of local field potentials of large neuronal populations, is the cross-frequency coupling between slow and fast frequencies. Such coupling has been suggested as a functional role in neural computation across different brain areas [49]. We explored whether this type of coupling occurs in GC oscillations and how they are regulated by the different mechanisms of synapses.

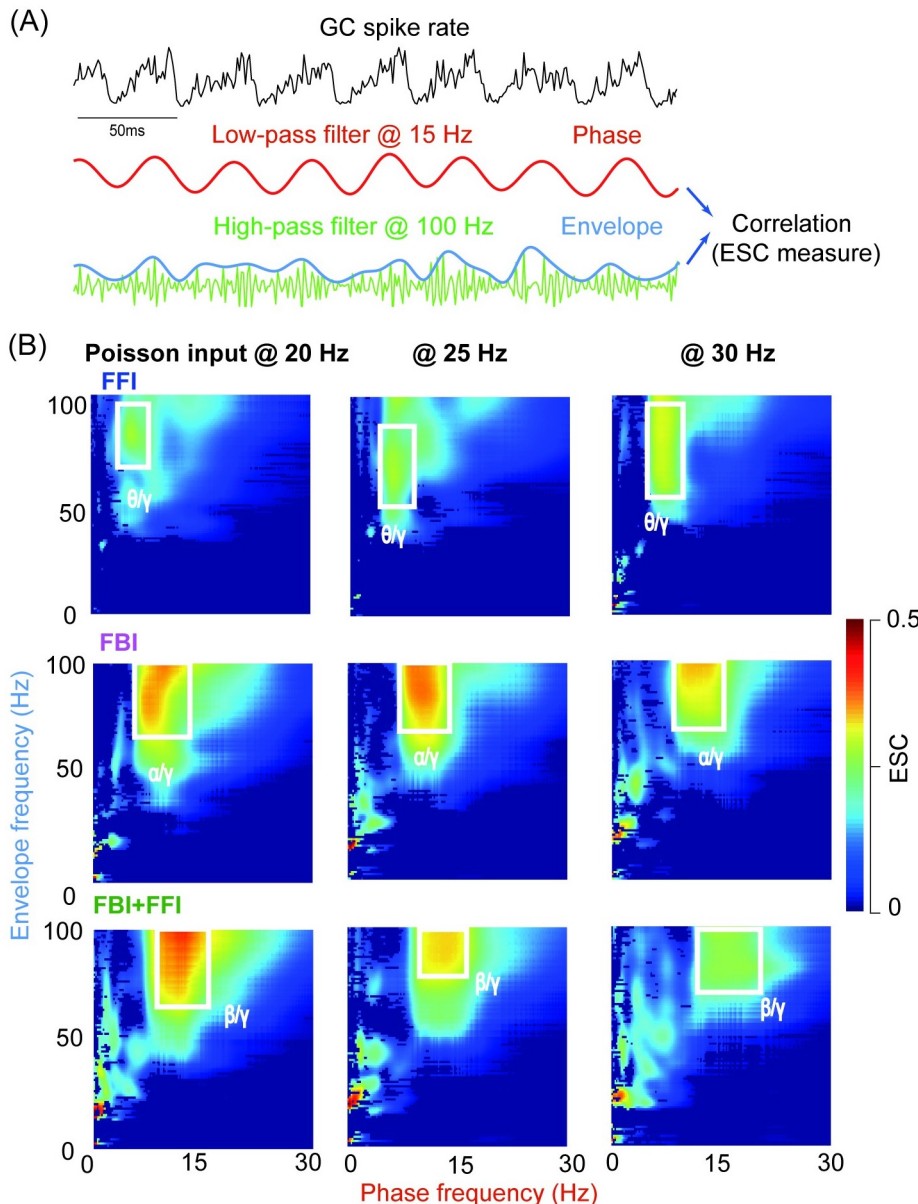

**Fig 5. Cross-frequency coupling of network oscillations induced by gap junction.** (A) Schematic view of phase-amplitude coupling. The raw signal of GC population spike rate with 25 Hz Poisson MF inputs in the FFI+FBI network. The low-pass filtered signal (middle, filtered at 15 Hz) for phase information. The high-pass filtered (bottom, filtered at 100 Hz) signal for amplitude envelope. The strength of phase-amplitude coupling was quantified as the envelope-signal coupling (ESC) correlation. (B) The ESC measure as a function of frequencies of phase (low-pass filter) and envelope (high-pass filter) computed for GC oscillations with Poisson inputs at 20, 25, and 30 Hz under three scenarios: FFI (top), FBI (middle), and FFI+FBI (bottom).

We quantified the coupling with phase-amplitude coupling (PAC) that was characterized by the correlation between the phase of low frequency and the amplitude of high frequency. The illustration of the approach is shown in Fig 5A, where the GC population spike rate was filtered with a pair of filters at different frequencies. The phase was obtained using a low-pass filter at 15 Hz, whereas the amplitude was given by the envelope of filtered signal using a high-pass filter at 100 Hz. Then the correlation coefficient was computed with this pair of signals as

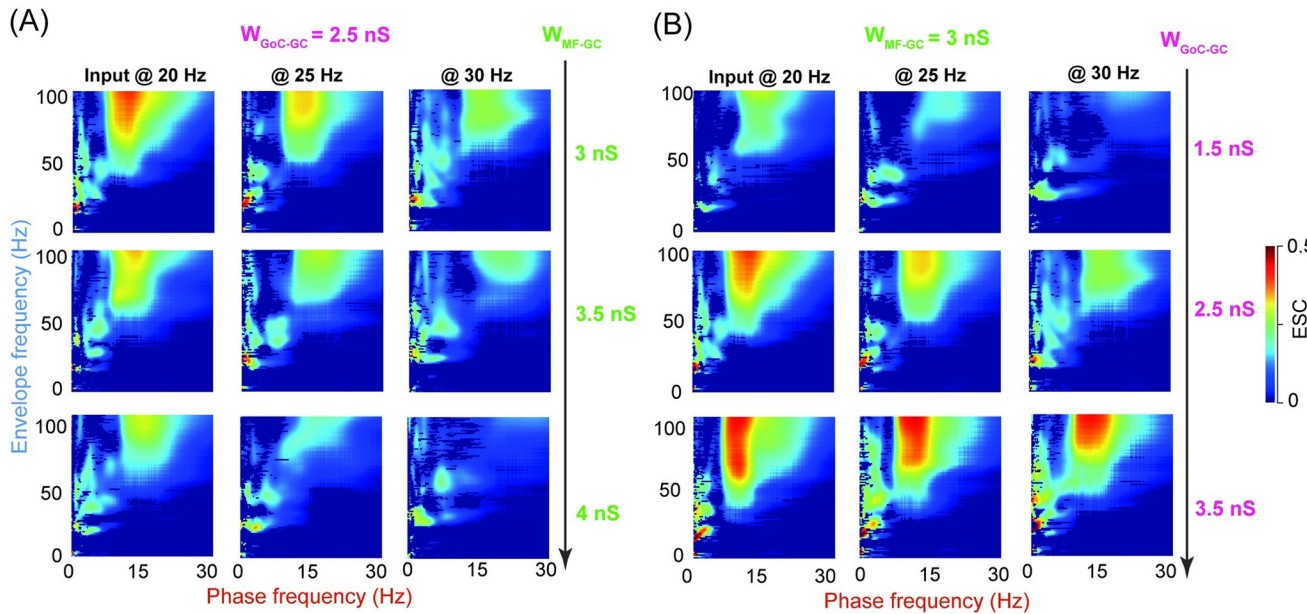

**Fig 6. Phase-amplitude coupling of GCs reinforced by GoC inhibition.** (A) The ESC measure as a function of frequencies of phase (low-pass filer) and envelope (high-pass filter) computed for GC oscillations with Poisson inputs at 20, 25, and 30 Hz in FFI+FBI networks. The change of the ESC over increasing $W_{MF-GC}$ excitation with fixed $W_{GoC-GC}$ inhibition. (B) Similar to (A), but for increasing $W_{GoC-GC}$ inhibition with fixed $W_{MF-GC}$ excitation.

the index for the PAC strength, named envelope-to-signal correlation (ESC) [45] (Fig 5A). Utilizing a wide range of frequencies for both low and high pass filters, we noticed there is a strong PAC shown shown by the PACgram [46] in Fig 5B with different Poisson inputs. With a specified input, PAC moves from low $\theta$ to high $\beta$ in phase frequency, whereas maintaining in the high $\gamma$ range in envelope frequency, as a consequence of network inhibition: FFI only gives PAC at $\beta/\gamma$, whereas FBI shifts PAC to $\alpha/\gamma$, and incorporating FBI and FFI together makes PAC at strong $\beta/\gamma$ range. As the input strength increases from 20 Hz to 30 Hz, PAC was reduced and narrowed in a smaller region of both low and high pass filter frequencies. Such a strong coupling can only be observed with gap junction in the network and greatly impaired by the short-term plasticity of GoC-GC synapses (S11 Fig).

As explained above, the balance of excitation and inhibition on GCs can change network oscillations, we further explored the effect of varying excitation and inhibition on PAC. When excitation $W_{MF-GC}$ increases (Fig 6A), PAC becomes weaker and suppressed with strong excitation. Interestingly, when inhibition $W_{GoC-GC}$ increases, PAC becomes stronger and more prominent (Fig 6B). These results suggest that inhibition plays a role in facilitating the PAC of GCs. In contrary to GCs, the PAC of GoCs increases when the input increase (S12 Fig). However, there is little PAC between GoCs and GCs (S12 Fig). Taken together, these results show that there is a strong PAC within the population of GCs and GoCs, respectively. Particularly, inhibition from GoCs plays a primary role in controlling PAC. Detailed network connections between GoCs and GCs, e.g., feedback inhibition, as well as gap junctions in GoCs, are noteworthy for promoting the PAC of GCs.

## Discussion

In this work, we present a network model with a unified description of the contribution of GoC inhibition to the synchronous oscillation of the GC population. Our work revealed the mechanisms of how GC oscillations are generated and shaped by different effects of inhibition,

including feedforward inhibition, feedback inhibition, short-term plasticity, and gap junction, from GoCs. With these biological details included, our model provides a testbed for the functional role of the granular network synchronous dynamics of the cerebellum in more general phenomena of neural oscillations existing in not only Purkinje cells, but other interacted brain areas through cortico-cerebellar communication. In line with the recent view of the novel roles of the cerebellum in cortex dynamics [50], our model is able to examine the different roles of GoC inhibition in neural oscillations by considering a wide range of experimental observations in the cerebellum and cortex.

## Network oscillation with different types of inhibition

Feedback inhibition has been suggested as a general mechanism in generating oscillation of neural networks [51, 52]. In the granular layer of the cerebellar cortex, previous studies have indicated that neural oscillations relate to the feedback loop between GoCs and GCs [19, 30, 35, 53]. Moreover, oscillations can be modulated by the balance of excitation and inhibition [33, 34] and feedforward inhibition [31, 32]. Consistent with these findings, our results indicate that feedback inhibition is crucial for generating oscillation that can be further modulated by the balance of excitation and inhibition inputs.

By analyzing excitatory connectivity in relation to GC oscillations, we found that synaptic excitation has different sensitivities to modulate oscillation. For instance, the strong strength of MF-GC synapses is required to maintain oscillations of GCs, whereas the weak strength of MF-GoC synapses allows broadening the emergence of oscillation. These results are consistent with earlier findings [19, 53]. Furthermore, the strength of GC-GoC synapses controls the oscillation and can increase the oscillation frequency of the network. However, for all changes of synaptic excitability, the oscillation frequency increases with increased excitability, but decreases with increased inhibition. It is in line with the observation that reducing GoC activity can stabilize high-frequency oscillations [54]. Overall, the oscillation frequency shows a linear relationship along the balanced excitation and inhibition inputs on GCs in the parameter space. Therefore, even for the same input, different combinations of excitation and inhibition can lead to a wide range of network oscillations. The cerebellum can maintain oscillation at different frequencies to complete different motor control, where both slow and fast GC oscillations have specific roles in cerebellar signal processing [17, 47, 54, 55]. Additionally, such a combination plays a functional role in gating signals for information processing and activity selection for general network dynamics [56–58].

## Gap junction and feedback inhibition in synchronous oscillations

Inhibitory chemical synapses promote oscillation by effective synaptic time constants and connectivity [59, 60], here we noted that introducing of short-term plasticity in GoC-GC synapses can significantly suppress GC oscillations. In contrast, the short-term plasticity of MF-GC synapses has little effect on GC oscillations. The oscillation dynamics has up and down states, in which the downstate is caused by inhibition input. With the STP in GoC-GC synapses, since the STP depends on the recent history of spiking activity, it can amplify asynchronous activity during the downstate, then dampen network oscillations. In many cortical regions, gap junctions between inhibitory interneurons have been suggested to contribute to network oscillations [26, 27, 29, 30, 61, 62]. We also found that oscillations generated by gap junctions are more robust and can overcome the dampening effect of the STP, in line with the findings that gap junctions can increase the power of oscillations [63].

Furthermore, the oscillation induced by gap junction is different from that driven by FBI. We observed that the oscillation frequency generated by FBI is higher than that by gap

junction. This may be due to that gap junctions between GoCs participate low-frequency oscillatory motor control in cerebellar processing [37]. In addition, we found that, in the existence of gap junction, the excitability of MFs to GCs facilitated oscillation, whereas increasing the strength of MFs to GoCs can depress oscillation. However, for the FBI, the paired FFI has little effect on the oscillation frequency. MF inputs increase the excitability of GoCs and cause GoCs to fire more easily, thus destroying the synchronization of GoCs. This observation is consistent with the dynamic role of gap junction, especially when MF firing frequency is low [30, 37, 64]. Our results suggest that gap junctions drive the oscillatory activity of GCs by synchronizing them with GoCs. Additionally, there are similarities between two ways of generating oscillations, such as, the oscillation frequency increase monotonically with the excitation level and is balanced by the inhibition through both mechanisms of FBI and gap junction.

## Cross-frequency coupling in neural oscillations

Electrophysiological recordings in animals, including humans, show that oscillatory activities are modulated in several frequency bands [65]. Recent findings suggest that cross-frequency coupling (CFC) plays a functional role in neuronal computation [49, 66, 67]. For instance, there is a strong correlation between CFC strength and performance in learning [68] and encoding of behaviors [69, 70]. It has been observed that fast oscillation is modulated by slow oscillation in different brain areas, such as hippocampus [71, 72], basal ganglia [71], neocortex [73], and cerebellum [74].

Underlying neural activities for CFC in Parkinson's disease have been suggested forming synchronized spikes in neural circuits, including the subthalamic nucleus [74]. Therefore, CFC in the cerebellum could play a role in motor control. Here, by analyzing CFC in the granular network with three different connections of network structure in terms of GoC inhibition, we revealed that individual network components could modulate CFC. Moreover, MF-GC synapses produced weak CFC whereas GoC-GC synapse facilitated CFC in GCs. It indicates that the GoC inhibition is important for the CFC in GCs. As GCs are the sole output of the granular layer of the cerebellum, CFC in GCs could potentially contribute to motor activities interacting with other subthalamic nucleus [74]. However, future studies are needed to explore how the CFC is propagated and regulated by downstream neurons of GCs, particularly Purkinje cells that are the output of the cerebellum and deliver a direct interaction with other parts of the brain. Together with diverse network scenarios, these building blocks of neural computation could greatly contribute to the role of the cerebellum in functions from the sensorimotor controller to cognition modulator [50].

## Implications for other systems

Our work here is consistent with other previous modeling work on synchrony and oscillation of neural networks. It is well observed in models that gap junction can synchronize neural spikes [26, 27, 75–77]. Experimental findings in the granular layer of the cerebellum suggest that gap junction can generate low-frequency (<30 Hz) oscillations in GoCs, which was well modeled using the data of gap junctions recorded from GoCs [37]. Here our model, based on this model of gap junction, but included coupled GCs, can explain the observation that GoCs deliver rhythmic inputs to GCs [37]. As a result, GC oscillations are mediated by GoCs, consistent with other modeling work [19]. Furthermore, our findings of tuning GC oscillations by different formats of GoC inhibition suggest that downstream Purkinje cells, that receive inputs from GCs as the output of the granular layer, could show coherent oscillations [19, 47]. Additional modulation sent out from Purkinje cells can further play a functional role in regulating synchronization of other brain areas [78, 79].

In this work, we used some genetic properties of neural networks, coupling between excitatory and inhibitory cells [33, 34], and ubiquitous features of many other neural systems, e.g., gap junction and feedback inhibition [22, 80]. Thus, the computational principles revealed in our work can provide new insights for studying other neural systems as well. Here we used a specific network topology of 2D grid space to easily take into account gap junctions, however, depending on the species and systems, other types of network topology can be adopted and modeled to realize oscillation behaviors [80, 81]. Nevertheless, real scenarios in neuronal systems employ laminar, column-like, and other designing principles for neural architectures [82]. These network scenarios enable us to implement efficient and devious types pf neural computations.

## Limitations

Here we only addressed synchronous oscillation of the granular layer of the cerebellum using a network model of GCs and GoCs installed with a variety of synaptic mechanisms. Besides short-term plasticity, another form of frequency-dependent plasticity is spike-timing-dependent plasticity that was found recently at the MF-GC synapses with a preferred frequency of 6–10 Hz [83]. This additional factor may also contribute to the synchronous oscillation of the GC population. The exact functional implication of this new finding remains to be elucidated by more detailed experimental and modeling studies.

It is also known that the granular layer shows the dynamics with diverse timescales and beyond the synchronized activity. These seemingly contradictory viewpoints are stemmed from the complex structure of the cerebellum that has been oversimplified traditionally [84]. The functions and dynamics generated by the complexity of the neural network structure are much more diverse in the cerebellum [50].

With an organization of roughly ten lobules in the cerebellum [85], different functions of the granular layer are also align with different lobules. One particular and well-studied example of multiple time-scale dynamics is unipolar brush cell [86], a relatively new cell type targeting to GCs. These cells, largely distributed in the lobules dedicated to the vestibular system, are suggested as internal mossy fibers or relay cells to diversify the time scale of external vestibular signals and form both excitatory and inhibitory dynamics over fast and slow time scales for Purkinje cell outputs [38, 87–89]. In addition, some types of GC inputs to GoCs were suggested to be inhibitory through the extra-synaptic spillover mechanism [90]. These diverse structure components of neurons and synapses were not included in this study. Future work is needed to take into account specific lobules for providing more fruitful insights into the role of different formats of GoC inhibition in the dynamics of the granular layer network.

## Supporting information

**S1 Fig. Related to Fig 1. Diverse firing patterns of GCs (top) and GoCs (bottom) induced by feedforward and feedback GoC inhibition**. (Left four panels) Characteristics of spike responses in three scenarios. Box plots represent quartiles (minimum, 25%, median, 75% and maximum values) for the mean of inter-spike intervals (ISI), standard deviation (SD), coefficient of variation (CV), and average spike rate, of 50 trials. Each trail (each data point) is a 10 second population firing rate averaged over all GCs or GoCs. The stimulus to the network is 50 Hz Poisson input. (Right panel) Neural responses are distinct across three scenarios visualized by principal component analysis (PCA) of population spikes of 50 trials. Black circles are the cluster centers and colored ellipse outlines indicate 95% confidence intervals calculated by the k-means clustering algorithm.
(TIF)

**S2 Fig. Related to Fig 2. The relationship of the GC firing rate and excitation (E) and inhibition (I)**. (A) Time courses of the membrane potential, MF-evoked EPSC ($V_{hold}$ = −70 $mv$), GoC-evoked IPSC ($V_{hold}$ = 0 $mv$) and E-I delay obtained as cross-correlations between EPSC and IPSC of one GC. The Poisson stimulus is at 25 Hz. $I_{delay} > 0$ means the I delay, otherwise $I_{delay} < 0$ as the E delay. (B) Time courses of GC firing rate, raster plots of $\delta$ of all GCs and the averaged $\delta$ over all GCs in two conditions of E/I = 1/1 (without oscillation) and E/I = 3/4 (with oscillation). (C) The GC population firing rate as a function of E-I delay averaged over all GCs, *i.e.*, the average of the rater plot in B. (D) Cross-correlations of firing rate and $\delta$. (E) The peaks of firing rate ($\hat{f}$) and amplitudes of oscillation ($\hat{\delta}$). (F) $\hat{f}$ as a function of $\hat{\delta}$ in different E-I combinations with FBI and FEI+FFI. Linear fits indicated by red solid lines.
(TIF)

**S3 Fig. Related to Fig 2.** The change of oscillation frequency (OF) under a range of weights in different synapses.
(TIF)

**S4 Fig. Related to Fig 3.** Spike time jitters of GCs changed by the GoC-GC synapses with STP (blue) or without STP (red).
(TIF)

**S5 Fig. Related to Fig 3. Network oscillation suppressed by short-term plasticity (STP) of GoC-GC synapses**. (A) Profiles of STP facilitation and depression controlled by the STP parameter U. Inhibitory postsynaptic potentials recorded at GCs induced by GoCs for both slow and fast synaptic components. (B) Suppression of network oscillation independent on the detailed profiles of STP, either facilitation or depression, for both components of fast and slow dynamics of GoC-GC synapses. Spike raster and population firing rate of all GCs and GoCs, and the corresponding power spectrum and auto-correlation (AC) of population firing rate. (C) Oscillation frequency (OF) suppressed by STP over a range of the parameter U in GoC-GC synapses. Networks with FBI.
(TIF)

**S6 Fig. Related to Fig 3.** (A) Modulation of the oscillation frequency (OF) over $W_{GC-GoC}$ and $W_{GoC-GC}$ strengths at different weights of $W_{MF-GC}$, under the conditions of presence and absence of the STP in MF-GC synapses. Here is the FBI network with Poisson inputs at 25Hz. The GoC-GC synaptic STP was included. (B) Similar to A, but the GoC-GC synaptic STP was turned off. The MF-GC STP has little effect on network oscillation, compared to the GoC-GC STP.
(TIF)

**S7 Fig. Related to Fig 1. GC responses with gap junction of GoCs in the network model**. (A) (Left) The layout of network model with gap junctions. A 2D grid showing the position of 144 GoCs (black dots) and 2000 GCs (red) in the model. 144 GoCs are arranged on a 12x12 grid with 33 $\mu$m spacing, where the location of each GoC was drawn randomly from the grid vertices, using a uniform distribution between ± 25% of grid spacing both in x and y directions. The radius of each GoC was drawn randomly from 0.7 to 1.3 times the average radius (70 $\mu$m) as well as its density of processes (from 0.7 to 1.3 relative to average). The extensions of two GoCs are represented by a green and cyan disk, respectively. The weight conductance between two cells was taken proportional to the area of overlap between the two disks and to the relative densities of processes of both cells. (Right) The distribution of the weights of all gap junctions as a function of the distance between GoCs. (B) Summary plots of metrics for a group of 50 trials in three network connections. Box plots represent quartiles (minimum, 25%,

median, 75% and maximum values) for mean ISI, standard deviation (SD), coefficient of variation (CV), spike number. Three clusters using two principal components (PAC) and the k-means clustering methods. Black circles are the cluster centers and ellipse perimeters indicate 95% confidence intervals. All colors correspond to three scenarios of GoC inhibition.
(TIF)

**S8 Fig. Related to Fig 4 and S7 Fig.** Network oscillation boosted by gap junction at different levels of gap junction strengths (top, reduced weight as 75% compared to the default case shown in S7 Fig; bottom, increased weight as 125%). Spike raster and population firing rate of GCs and GoCs, and the corresponding power spectrum and auto-correlation (AC) of population firing rate in three network scenarios of FFI, FBI, and FBI+FFI. GoC-GC STP included.
(TIF)

**S9 Fig. Related to Fig 4. Network oscillation with gap junctions in GoCs, but without GoC-CC synaptic STP**. (A) The spike raster and population firing rate of GCs and GoCs (left), and the corresponding power spectrum and autocorrelation (AC) of population firing rate (right), in three network scenarios of feedforward, feedback and both types of inhibition. (B) Network oscillation metrics characterized by the amplitude (Amp), inter-event interval (IEI) and width. Histograms of three metrics for GC (top) and GoC (bottom).
(TIF)

**S10 Fig. Related to Fig 4 and S3 Fig.** The change of oscillation frequency (OF) with different synapses, under the condition of gap junction included.
(TIF)

**S11 Fig. Related to Fig 5B.** Cross-frequency coupling of network oscillations impaired by STP when gap junctions are not included. The ESC measure as a function of frequencies of phase (low-pass filter) and envelope (high-pass filter) computed for GC oscillations with Poisson inputs at 20, 25, and 30 Hz under the scenario of FBI. Different profiles, facilitation or depression controlled by the parameter U, of STP have similar effects on ESC.
(TIF)

**S12 Fig. Related to Fig 6. The Cross-frequency coupling in the population of GoCs and between GoCs and GCs**. (A) The strength of phase-amplitude coupling (PAC) quantified for GoCs oscillations with different Poisson inputs at 20, 25, and 30 Hz under three types of network connections. (B) Same as in (A), but for PAC between GCs and GoCs. (Left) PAC between GCs (phase) and GoCs (envelope). (Right) PAC between GoCs (phase) and GCs (envelope).
(TIF)

## Author Contributions

**Conceptualization:** Lingling An, Jian K. Liu.

**Data curation:** Yuanhong Tang, Jian K. Liu.

**Formal analysis:** Yuanhong Tang.

**Funding acquisition:** Lingling An, Quan Wang, Jian K. Liu.

**Investigation:** Yuanhong Tang, Lingling An, Jian K. Liu.

**Methodology:** Yuanhong Tang, Lingling An, Jian K. Liu.

**Project administration:** Lingling An, Jian K. Liu.

**Resources:** Lingling An, Jian K. Liu.

**Software:** Yuanhong Tang, Jian K. Liu.

**Supervision:** Lingling An, Quan Wang, Jian K. Liu.

**Validation:** Yuanhong Tang, Jian K. Liu.

**Visualization:** Yuanhong Tang, Jian K. Liu.

**Writing – original draft:** Yuanhong Tang, Lingling An, Jian K. Liu.

**Writing – review & editing:** Yuanhong Tang, Lingling An, Jian K. Liu.

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
