## [Decision Letter · Decision Letter 0]

25 Feb 2021

Dear Dr An,

Thank you very much for submitting your manuscript "Regulating oscillations of cerebellar granule cells by different types of inhibition" for consideration at PLOS Computational Biology.

As with all papers reviewed by the journal, your manuscript was reviewed by members of the editorial board and by several independent reviewers. In light of the reviews (below this email), we would like to invite the resubmission of a significantly-revised version that takes into account the reviewers' comments.

We cannot make any decision about publication until we have seen the revised manuscript and your response to the reviewers' comments. Your revised manuscript is also likely to be sent to reviewers for further evaluation.

Sincerely,

Michele Migliore

Associate Editor

PLOS Computational Biology

Lyle Graham

Deputy Editor

PLOS Computational Biology

Reviewer's Responses to Questions

**Comments to the Authors:**

Reviewer #1: inhibitory cells and gap junctions play a crucial role for many forms of neural oscillations – that is long known – but the detailed synaptic mechanisms are often not clear. The authors present a network model to test several excitatory/inhibitory conditions between mossy fibres, Golgi and granular cells in the cerebellum. They showed that feedback inhibition between Golgi and Granular cell is important to establish network oscillations, whereas gap junctions between Golgi cells maintain robust oscillations. Overall, the results are sound and the findings interesting, but the manuscript needs to address some urgent points, especially the novelty point, to proceed.

1) The main experimental data and major parts for the computational cell and network model are based on Dugue et al, 2009. The same paper also states the importance of gap junction coupled Golgi cells for oscillation. The here submitted manuscript needs to clarify where the novelty is compared to the Dugue work.

2) The authors do not provide experimental work alongside with the theoretical work to validate the results. Instead, they used already published experimental data to setup the model framework (see above). In my opinion it would be good to provide more evidence the findings by comparing to other models or systems. For example, the oscillations in the retina are well studied.

3) I was astonished that STP does not have a stronger effect on the oscillations. Spike-Timing Dependent Plasticity was shown in several papers to have a profound effect on oscillations at least in a feed forward model (Luz and Shamir 2016, PloS Comp Biol). I feel you should extend the discussion on that topic.

4) Can your model inflict seizure like oscillations? Or better asked: which parameter/condition needs to be changed to go ictal? That would be interesting for a wider audience.

Reviewer #2: The authors present modeling results addressing several interrelated questions concerning oscillations and synchrony in the cerebellum: how do feedback and feedforward Golgi cell inhibition differentially contribute to synchrony? How do gap junctions and chemical synaptic dynamics affect synchrony? And does the resulting oscillatory network state exhibit cross-frequency coupling?

While these are all interesting and relevant questions, and the modeling work shows potential, it fails to provide broader insights or conclusions regarding these phenomena. The conclusion “the interaction of various types of inhibition plays a crucial role in regulating synchronous oscillations of neural populations” is vague and likely true for essentially any system. While there are some more specific conclusions, they are essentially of the form “this phenomena was observed in this specific parameter range,” which fails to illuminate either mechanisms or functional relevance. These conclusions could be expanded into broader, impactful insights by several paths:

• Looking for explanations underlying the network-level trends: Can the effects of varying different coupling strengths be understood with a simple linear interaction model? How exactly does the STP desynchronize? What is the mechanism for cross-frequency coupling?

• Expanding on the functional relevance of the network synchrony by, for instance, linking the model to adaptation of Purkinje cell output. (See comment below on reconciling the synchronized vs diverse-timescale views of granule layer function)

• Connect the modeling work more directly to the past modeling work it builds on by more precisely identifying what questions were left open and how this work resolves those questions. This may have been intended, but wasn’t clear to me. This might require more precisely matching aspects of the model to the relevant previous work.

I also briefly note several specific issues found in the manuscript that are less comprehensive, but still critical to address.

• L78: “excitation and inhibition have contrary effects on the oscillation frequency, which rises with increasing excitability, and decreases as increasing inhibition” – this in particular is a conclusion that seems possibly explainable by considering the oscillation frequency of a simple linear neural mass model of the two coupled populations.

• L92: “[UBCs] operate like intrinsic MFs within the granular layer to diversify inputs and can be represented as phase shifted MFs [36],” – One of the primary references for previous modeling that this builds on is on GC-UBC interactions, in which the presence of new timescales/phase shifts of activity in the UBCs, not found in MF input, is shown to be important. Yet the authors claim modeling all MF and UBC input as Poisson spiketrains with the same properties is sufficient. This decision should be better justified, or they should be included in the model somehow.

• L112: the reason for analyzing the network data by k-means clustering is unclear. While this does emphasize the differences in activity between the three forms of inhibition, clustering is more appropriate if the identity of the groups are not known in advance. The same view of the activity space could be used without clustering, as the distinctions are quite clear by eye (although what form of data exactly PCA is applied to should be clarified).

• L122 “potential function for enhancing sensory representation and facilitating pattern separation 122 for downstream cells in the cerebellum.” - In distinct areas of work on the cerebellum, researchers may view the ideal functional state of the granular layer network as synchronized to enable effective motor control (e.g. the work on Golgi gap junctions this builds on) or as containing a diverse range of timescales/patterns to enable learning of precise timing (e.g. the work on UBCs this builds on). This work seems to only consider the former, but since it builds on both views and describes the conclusions as relevant for the function of the network as a whole, they should at least discuss this contradiction. Would a real network operate with multiple subnetworks that are each synchronized, or by generating a range of patterns on top of a globally synchronized ‘clock’ frequency?

• The authors in many places refer to gap junctions as providing inhibition or “a balance of excitation and inhibition” – that is maybe technically true, but neglects the simpler intuitive explanation that gap junctions typically synchronize by providing diffusive coupling (the effects act to cancel any difference in voltage)

• The authors consistently use the term ‘oscillations’ to refer only to population-level synchronized oscillations (as would be experimentally observable via LFP etc). It would add clarity to distinguish in which scenarios or parameter regimes there may be individual elements in the network with oscillatory activity that is not synchronized. See L68: “gap junctions between GoCs can also generate oscillations [26], [32], [34], [35].” – in some of these references, gap junctions are not exactly generating oscillations, but rather synchronizing single-cell oscillations to enable network oscillation.

• In the discussion, the authors often introduce experimental citations as explaining the modeling results, when really they should strive for the reverse, using the modeling to explain those experimental results. See L285: “the oscillation frequency increases with increased excitability, but decreases with increased inhibition, which may be due to that reducing GoC activity can stabilize high-frequency oscillations [45].”

• While the methods are clearly written in regards to introducing modeling details, it would be helpful to summarize certain aspects of the model choices in the main text, particularly where model assumptions are likely critical to the conclusions. For example, the presence and sources of heterogeneity are critical when discussing synchrony, but that detail is hidden deep in a methods table (I think heterogeneity is introduced only through threshold variability?)

• The authors should address whether they plan to release source code for their simulations for transparency and reproducibility. Building the simulations using one of the many widely used neural simulation packages would be additionally appreciated, as it simplifies reuse. If there were strong reasons not to do so, they might be worth mentioning.

• Numerous minor errors with word usage and sentence structure interfered with my reading of the manuscript, such that it's possible misunderstanding contributing to some of the issues raised above. I would advise the authors to get editing help from someone with full professional proficiency in English. A few examples in the author summary: “In the network” without introducing which network; “Feedforward inhibition is showed as”; “potentially for its paradigm-shifted role…”.

Reviewer #3: Comments are uploaded as an attachment.

**Have all data underlying the figures and results presented in the manuscript been provided?**

Reviewer #1: Yes

Reviewer #2: None

Reviewer #3: Yes

PLOS authors have the option to publish the peer review history of their article (what does this mean?). If published, this will include your full peer review and any attached files.

Reviewer #1: **Yes: **Gerrit Hilgen

Reviewer #2: **Yes: **Thomas Chartrand

Reviewer #3: **Yes: **Sheng-hong Chen
---

## [Decision Letter · Decision Letter 1]

27 May 2021

Dear Dr An,

Thank you very much for submitting your manuscript "Regulating oscillations of cerebellar granule cells by different types of inhibition" for consideration at PLOS Computational Biology.

As with all papers reviewed by the journal, your manuscript was reviewed by members of the editorial board and by several independent reviewers. In light of the reviews (below this email), we would like to invite the resubmission of a significantly-revised version that takes into account the reviewers' comments.

Given the widely different comments received for the original submission, it has been decided to include one more reviewer.

For this revised version, reviewers 2 and 4 still raise important issues on the way in which the authors present and discuss their model and findings.

In this last round of review, the authors should adequately address these issues.

We cannot make any decision about publication until we have seen the revised manuscript and your response to the reviewers' comments. Your revised manuscript is also likely to be sent to reviewers for further evaluation.

Sincerely,

Michele Migliore

Associate Editor

PLOS Computational Biology

Lyle Graham

Deputy Editor

PLOS Computational Biology

Given the widely different comments received for the original submission, it has been decided to include one more reviewer.

For this revised version, reviewers 2 and 4 still raise important issues on the way in which the authors present and discuss their model and findings.

In this last round of review, the authors should adequately address these issues.

Reviewer's Responses to Questions

**Comments to the Authors:**

Reviewer #1: The manuscript has significant improved. No objections from my side. Good to go!

Reviewer #2: (Relating to A-I-1, A-II-2, and overall novelty) – The revisions have helped clarify the distinction of the modeling methods from previous models, but still don’t provide clear arguments that the new features of the network shown here (interaction between GoC and GC for stronger synchronized oscillations despite STP) are either a more realistic description of experimental observations or essential to a proposed function. What biological insight does this added complexity gain us?

(A-II-1) – Although the additions are appreciated, they simply add complexity when the goal here should be to reduce it by teasing apart mechanisms, which I don’t see progress on. Indeed, in (A-II-4) the authors seem to agree that a simple explanation is possible for the e/I effects on oscillation frequency. To me, that should be a reason to simplify the manuscript by passing over such results with a quick explanation and focus on the more novel results. Another place where I imagine the results could be considerably simplified with more careful analysis is in aligning the results on cross-frequency coupling with the results on oscillation frequency and strength when the same parameters are varied – do the coupling results directly follow the underlying oscillation in an intuitive way?

(Fig. S5, new issue) – The authors again add quite a bit of new complexity to show the robustness of the demonstrated suppression of oscillations for both depression and facilitation, but should more clearly explain this counterintuitive result, or focus on a single set of experimentally supported STP parameters. It appears that the least suppression is observed for small U, but that is not explained. Also, how much is explained simply by varying the mean value of u, and what is u fixed at for the no-STP case?

Minor points:

(A-I-3) – There seems to have been some confusion here between reviewers and authors between STP and STDP. While STDP is certainly out of scope, that should probably be clearly stated in the manuscript to avoid confusion.

Reviewer #3: The authors have addressed most of the questions I raised. The major point2 will be further investigated in their future studies.

Reviewer #4: In this manuscript, Tang et al present a model for cerebellar oscillatory regulation via feedback inhibition of Golgi cells to granular cells. Their model predicts that the strength of the synapses between Mossy fibers and GOCs or GCs, presence and strength of GOC gap junctions and synaptic plasticity all work to modulate the amplitude and frequency of GC output.

This revised manuscript presents a detailed description of the model and responds to several of the prior comments. It falls short however of successfully answering other shortcomings previously mentioned, reducing the enthusiasm for an otherwise very interesting paper:

- The network model presented here is solely based on a prior report from a different group (Dugue et al Neuron 2009). It was suggested to compare it to other models or systems. This has not been done and remains a critical point. Is this model robust enough that the conclusions remain true if based on other experimental data?

- The term ‘oscillations’ remain confusing (e.g., see former Q-II-9 comment). While the fact that these are meant always to be synchronous oscillations may be obvious for readers from a narrow field, it is important to make this difference and the paper’s conclusions clear to a wider audience. A few sentences in the intro or results section should suffice.

- Q-II-11 remains a concern - a better explanation of the model and the assumptions clearly explained at the beginning of the results section is needed, especially for a wider audience. The details remain scant in the main text and the methods section will be too intimidating for a non-computational biologist. This will reduce the ability of many readers to interpret this model and limit the impact of the paper.

**Have the authors made all data and (if applicable) computational code underlying the findings in their manuscript fully available?**

Reviewer #1: Yes

Reviewer #2: **No: **There is no code found at the stated address, https://github.com/jiankliu/GC-CoC-Network

Reviewer #3: Yes

Reviewer #4: Yes

PLOS authors have the option to publish the peer review history of their article (what does this mean?). If published, this will include your full peer review and any attached files.

Reviewer #1: **Yes: **Gerrit Hilgen

Reviewer #2: **Yes: **Thomas Chartrand

Reviewer #3: **Yes: **Sheng-hong Chen

Reviewer #4: No
---

## [Editor Report · Decision Letter 2]

8 Jun 2021

Dear Dr An,

We are pleased to inform you that your manuscript 'Regulating synchronous oscillations of cerebellar granule cells by different types of inhibition' has been provisionally accepted for publication in PLOS Computational Biology.

Best regards,

Michele Migliore

Associate Editor

PLOS Computational Biology

Lyle Graham

Deputy Editor

PLOS Computational Biology

---

## [Editor Report · Acceptance letter]

23 Jun 2021

PCOMPBIOL-D-21-00018R2 

Regulating synchronous oscillations of cerebellar granule cells by different types of inhibition

Dear Dr An,

I am pleased to inform you that your manuscript has been formally accepted for publication in PLOS Computational Biology. Your manuscript is now with our production department and you will be notified of the publication date in due course.

With kind regards,

Katalin Szabo
